# Using a State-Bounding Observer to Predict the Guaranteed Limits of Drug Amounts in Rats after Oral Administration Based on an Uncertain Pharmacokinetic Model

**DOI:** 10.3390/pharmaceutics14040861

**Published:** 2022-04-14

**Authors:** Zuzana Vitková, Martin Dodek, Jarmila Pavlovičová, Anton Vitko

**Affiliations:** Faculty of Electrical Engineering and Information Technology, Institute of Robotics and Cybernetics, Slovak University of Technology in Bratislava, Ilkovičova 3, 812 19 Bratislava, Slovakia; zuzana.vitkova@stuba.sk (Z.V.); jarmila.pavlovicova@stuba.sk (J.P.); anton.vitko@stuba.sk (A.V.)

**Keywords:** uncertain pharmacokinetic model, state observer, guaranteed concentrations

## Abstract

In the first part of this paper, the problem of using an uncertain pharmacokinetic model is resolved to determine drug concentrations in rats after the oral administration of drug suspensions with and without added tenside. To this end, a generalized pharmacokinetic model determining the guaranteed limits of drug concentrations was designed. Based on this, the design of the so-called state-bounding observer is described in the second part. Rather than being driven by the output of the pharmacokinetic model, the observer can be driven exclusively by a concentration collected from a suitable part of the body and predict the possible risk of the drug concentration not remaining within the therapeutic range for a sufficiently long time. Specifically, the observer determines the upper and lower limits of the concentrations in all the compartments, especially those that are inaccessible for the collection of samples. The proposed approaches are demonstrated by examples.

## 1. Introduction

The processes running in a living system are dynamic in the sense that their future behavior is, under normal conditions, exactly determined by the state of the system and the input that is fed into it. Therefore, many theories that were originally developed for the analysis and control of artificial processes in classical engineering may be applied to living systems as well.

Contemporary biocybernetics successfully resolves many problems in biology and medicine. The need to improve drug therapy calls for the analysis of macro and micro bioprocesses at more abstract levels. This makes it possible to extract knowledge and useful information from in vivo experiments and to synthesize appropriate abstract models based on these data.

However, the evaluation of the traditional integral characteristics of drug kinetics, such as, clearance (CL), half-life (t_1/2_), area under the curve (AUC), bioavailability (F), mean residence time (MRT), mean absorption time (MAT), mean variation time (MVT), and others, is often not sufficient. Both drug designers and doctors need more detailed knowledge of the drug exchange process between various parts of the body.

This problem becomes even more involved if, due to the actions of various endogenous and exogenous factors, the pharmacokinetic parameters vary even within 10% of their nominal values. Significant influences also come from the addition of constituents in dosage form. The most frequently used constituents are tensides such as monolaurin, monopalmitan, and monostearan of sucrose.

Consequently, the pharmacokinetic parameters identified from the results of in vivo measurements should be considered as relatively rough approximations of their true values. The only option open to the drug designer is to assess the intervals within which the pharmacokinetic parameters vary. This knowledge is undoubtedly of the utmost importance to provide safe therapy, because drug concentrations must be kept within an acceptable therapeutic range regardless of the current state of the body. If the concentration exceeds the therapeutic range, it either acts as a toxic agent or becomes ineffective.

It is known [1] that after oral administration, drug absorption from the gastrointestinal tract (GIT) can be affected by the addition of tensides. If the concentration is below the critical micellar concentration (CMC), the tenside increases the rate of absorption, while if the concentration is above the CMC, then it decreases this rate [2,3,4]. Simultaneously with drug absorption, drugs are excreted from the body. Therefore, it might be inappropriate to evaluate the rate of absorption by simply measuring the change in drug concentration in the blood. Hypothetically, one way to obtain information on drug concentrations in various parts of the body could be to implement an appropriate sensor in each compartment of the body. Clearly, this is impossible both from an ethical and from a technical perspective.

However, a more feasible way is to implement a sensor of the drug concentration in only one part of the body, e.g., in the peripheral compartment, and then exploit this information to estimate the current concentrations in the rest of the compartments, which is also commonly known as the state observation problem [5,6,7,8]. In particular, we focus on the problem of determining the limiting values within which the actual concentrations in the compartments are maintained, despite the uncertain pharmacokinetics involved. The aim of this paper is two-fold. In the first step, the influence of the tenside-monolaurin of sucrose (MLS) on the amount of sulfathiazole in rats is analyzed. This problem was originally resolved in [1], but under the assumption of a pharmacokinetic model free of parametric uncertainties.

In this paper, we examine a similar problem, but under conditions of uncertain pharmacokinetic parameters. Because this situation is commonly encountered in the life sciences, there is a need to estimate at least the intervals within which the true values of the drug amounts in various compartments may be obtained. To solve this problem, a numeric algorithm, known in systems theory as the observer, is proposed.

In the second step, the designed observer is used as a predictor of the intervals within which the drug amounts can be expected, while measuring the drug amounts in a single site.

## 2. Materials and Methods

The materials and methods are described in detail in [1], so here we summarize only the basic facts. The in vivo experiment consisted in the administration of sulfathiazole suspensions with and without tenside (MLS) to rats. The drug without tenside was administered first and the corresponding series of concentration samples was recorded. Subsequently, the same process was repeated, but with added tenside. Based on the obtained concentrations, a parsimonious structure of the two-compartment pharmacokinetic model was designed. Subsequently, this model was analyzed for its parametric identifiability and then, finally, identified. The methodology used in [1] can be applied equally well to all oral dosage forms, including those with controlled release and even sophisticated drug delivery systems (DDS).

Note that in [1], the model parameters were not considered uncertain. The importance of resolving the problem outlined in the title is stressed by the fact that drug concentrations in all compartments should be known to design effective dosing regimens, while these concentrations cannot be directly measured in vivo.

### 2.1. Need for a State-Bounding Observer

In the treatment of various diseases, physicians need to know the drug concentrations in the relevant organs (compartments) of the body. However, since some of these cannot be measured, they have to be estimated by a special system: the observer.

The observer is a numeric algorithm that “observes” the measured drug amount in a chosen compartment and generates an estimate of the drug amount in the remaining compartments. From the system’s point of view, the state-bounding observer is an additional tool processing the output of the pharmacokinetic model.

For all the compartments, two independent observers, each producing a limiting curve, are designed. The limiting curves delineate the area from which the drug concentration cannot “escape”, even if the values of the pharmacokinetic parameters are estimated with limited accuracy. Resolving the state estimation problem is especially important for compartments that are not available (for technical, medical, or ethical reasons) through the collection of blood samples. These involve, for instance, the gastrointestinal tract (GIT) and the veins.

Each observer is driven by the instantaneous drug concentration in the output compartment of the pharmacokinetic model, i.e., the compartment from which the blood samples are collected. This output compartment is commonly chosen as the most suitable for sample collection.

The derivation of state-bounding observers for typical bio-systems is hampered by the fact that characteristic quantities must take exclusively non-negative values. These, involve, for example, the drug concentration, drug amount, blood pressure, body temperature, etc. These types of system belong to the category of “positive systems”, of which compartmental pharmacokinetic models are a subset [6].

The requirement for system positivity makes the derivation of equations describing “positive” state-bounding observers rather a demanding problem. For this reason, any mathematical rigor is omitted, so only the equations that are essential to understanding the main ideas are presented. For interested readers, we reveal that the derivations are commonly based on the theory of linear programming (LP) or linear matrix inequalities (LMI) [7,8]. The approach adopted in this paper follows the theory presented in [9,10,11,12,13,14,15].

### 2.2. Uncertain Compartmental Model

Perhaps the most difficult problem one may encounter while trying to predict the drug concentrations in body compartments is the lack of an exact and reliable pharmacokinetic model. Pharmacists know that the model structure is typically chosen on the basis of the properties of the drug at hand, the method of administration, the required level of model comprehensiveness, and last, but not least, the experience of the researcher. Clearly, the structure must be as parsimonious as possible, while its in silico behavior should reliably reflect the observations of real processes.

This paper presents two uncertain compartmental pharmacokinetic models. The first is intended to explain how an uncertain model can be derived. The second is proposed to demonstrate a more practical application of the observer’s prediction of whether the drug concentration can potentially extend beyond the therapeutic range if there is a mismatch between the living body and its model. It should be mentioned that the first model was already identified in our recent work [1], whereas the second proposed model is identified in this paper.

There are numerous reasons why the parameters of pharmacokinetic models cannot be determined exactly and with high confidence. First, there is a relatively small number of blood samples that can be collected compared to the number of parameters that can be identified, which makes the identification problem underdetermined and poorly conditioned. In addition, the sensors feature limited accuracy, the numerical computations feature rounding errors, the conditions in which the collection of samples is carried out are variable, etc. These factors cannot be simply neglected, so parametric uncertainties should be included in both the pharmacokinetic model and the state-observer algorithm.

The blood volume of rats can be easily calculated, as reported in [16]. The administered dose of suspension in [1] was expressed in milligrams, so the concentrations were converted to amounts. Accordingly, all further variables will be presented in terms of drug amounts rather than using their concentrations.

### 2.3. Nominal and Uncertain Pharmacokinetic Models

These two notions are represented by the two-compartment model with a single input and a single output (SISO) depicted in Figure 1. The dynamics of the model are described by the set of linear differential equations (1). The state of the system represents the set of instantaneous drug amounts in the individual compartments, so the state vector x comprises n components as the number of compartments.


u is the system input—an instantaneously administered dose $M_{0}$;$k_{e1}$, $k_{e2}$ are the rate constants of the drug elimination;$k_{a}$ is the absorption-rate constant;$x_{1}\left(t\right)$ is the time-dependent drug amount in the GIT;$x_{2}\left(t\right)$ is the time-dependent drug amount in the central compartment—the blood;y(t) is the system output—the observed (measurable) drug amount equal to $x_{2}\left(t\right)$.


The equations of the nominal compartmental model are defined as follows:(1)$$\begin{array}{l} \underset{\dot{x}}{\underbrace{\left(\begin{array}{c} {\dot{x}}_{1}\\ {\dot{x}}_{2} \end{array}\right)}}=\underset{A}{\underbrace{\left[\begin{array}{cc} -\left(k_{a}+k_{e1}\right) & 0\\ k_{a} & -k_{e2} \end{array}\right]}}\underset{x}{\underbrace{\left(\begin{array}{c} x_{1}\\ x_{2} \end{array}\right)}}+\underset{b}{\underbrace{\left(\begin{array}{c} 1\\ 0 \end{array}\right)}}u\left(t\right)\\ y=\underset{c^{T}}{\underbrace{\left(0,\;1\right)}}\underset{x}{\underbrace{\left(\begin{array}{c} x_{1}\\ x_{2} \end{array}\right)}} \end{array}$$
where the symbol A denotes the system matrix, i.e., the relation between the state vector x and its time derivatives, and b and c are column vectors, known as the control and the observation vectors, respectively [5,7]. The symbol $c^{T}$ means the transposition operation of the column vector c; hence, $c^{T}$ is a row vector. As mentioned above, the parameters of model (1) were identified from the results of the in vivo experiments performed on the rats. The aim of our earlier research was to analyze the effect of the tenside on the absorption-rate constant. The methods and materials used in the experiment are discussed in detail in [1], while the crucial results of the in vivo measurements are summarized in Table 1.

Based on these results, the following values of the model parameters were identified [1]. Before the addition of the tenside:(2)$$k_{a}=0.030272{\;h}^{-1}\;k_{e1}=0.649656{\;h}^{-1}\;k_{e2}=0.338477{\;h}^{-1}$$

After addition of the tenside:(3)$$k_{a}=0.045815{\;h}^{-1}\;k_{e1}=0.959041{\;h}^{-1}\;k_{e2}=0.242913{\;h}^{-1}$$

It is obvious that the administered dose 50 mg is always known exactly, but we do not know the drug amount that enters the GIT just after the oral administration. Therefore, this value should be considered as an uncertain quantity, namely $x_{1}\left(0\right)\approx 50\;\mathrm{mg}$. The model parameters are also uncertain because they are identified with finite accuracy and limited confidence. The uncertainties of the parameters are represented by the uncertain matrix ΔA  and the vector Δc. A general form of the uncertain pharmacokinetic model is given as follows:(4)$$\dot{x}\left(t\right)=\left(A\pm \Delta A\right)x\left(t\right)\;y\left(t\right)={\left(c\pm \Delta c\right)}^{T}x\left(t\right)\;x\left(0\right)-\Delta x\left(0\right)\leq x\left(0\right)\leq x\left(0\right)+\Delta x\left(0\right)$$
where 0≤Δ ≤1 is a fraction by which the corresponding quantity can change.

In particular, the uncertain model corresponding to (1) takes the following form:(5)$$\begin{array}{l} \underset{\dot{x}}{\underbrace{\left(\begin{array}{c} {\dot{x}}_{1}\\ {\dot{x}}_{2} \end{array}\right)}}=\underset{A}{\underbrace{\left[\begin{array}{cc} -\left[\left(k_{a}+k_{e1}\right)\pm \Delta \left(k_{a}+k_{e1}\right)\right] & 0\\ k_{a}\pm \Delta k_{a} & -(k_{e2}\pm \Delta k_{e2}) \end{array}\right]}}\underset{x}{\underbrace{\left(\begin{array}{c} x_{1}\\ x_{2} \end{array}\right)}}\\ y=\underset{c^{T}}{\underbrace{\left(\begin{array}{cc} 0 & 1\pm \Delta \end{array}\right)}}\underset{x}{\underbrace{\left(\begin{array}{c} x_{1}\\ x_{2} \end{array}\right)}} \end{array}$$

Substituting the identified parameter values (2) into (1) yields Equation (6), which describes the nominal model:(6)$$\left(\begin{array}{c} {\dot{x}}_{1}\\ {\dot{x}}_{2} \end{array}\right)=\left(\begin{array}{cc} -0.6799 & 0\\ \;0.0303 & 0.3385 \end{array}\right)\left(\begin{array}{c} x_{1}\\ x_{2} \end{array}\right)\;y=\underset{c^{T}}{\underbrace{\left(\begin{array}{cc} 0 & 1 \end{array}\right)}}\underset{x}{\underbrace{\left(\begin{array}{c} x_{1}\\ x_{2} \end{array}\right)}}\;x\left(0\right)=\left(\begin{array}{c} 50\\ 0 \end{array}\right)$$

Assuming that the parametric uncertainties do not exceed 10% of their nominal values (Δ=0.1), according to (4), the upper and lower matrices $\underline{A}$, $\bar{A}$ and the vectors $\underline{c}$, $\bar{c}$ corresponding to nominal A and c are equal to:(7)$$\bar{A}=\left(\begin{array}{cc} -0.6119\; & 0\\ 0.0333 & -0.3046 \end{array}\right)\;\underline{A}=\left(\begin{array}{cc} -7.479 & 0\\ 0.0272 & -0.3723 \end{array}\right)\;\bar{c}=\left(\begin{array}{cc} 0 & 1.1 \end{array}\right)\underline{c}=\left(\begin{array}{cc} 0 & 0.9 \end{array}\right)\;x_{U}\left(0\right)=\left(\begin{array}{c} 50\\ 0 \end{array}\right)x_{L}\left(0\right)=\left(\begin{array}{c} 50\\ 0 \end{array}\right)$$

Let $x_{U}$ and $x_{L}$ represent the upper and lower limits that the drug amounts x may reach due to the uncertain pharmacokinetics. The numeric solutions of Equations (1) and (5) for the nominal amount x, for the upper amount $x_{U}\left(A=\bar{A}\right)$, and for the lower amount $x_{L}\left(A=\underline{A}\right)$, corresponding to the drug without the tenside and the parametric uncertainty Δ=±10%, are shown in Figure 2. The regions between the couples of curves $x_{1U}$, $x_{1L}$, and $x_{2U}$, $x_{2L}$ represent possible variations in the drug amounts $x_{1}$ and $x_{2}$ in the corresponding compartments. As can be observed, the uncertain values of parameters (2), (3) virtually do not affect the amount $x_{1}$ in the GIT, but the amount $x_{2}$ in the central compartment is significantly affected.

After the addition of the tenside, the second set of pharmacokinetic parameters was identified (see Equation (3)). The corresponding numeric solutions for the states in Figure 3 are quite different.

As shown in Figure 2a and Figure 3a, it is evident that the difference between the limiting amounts $x_{U}$ and $x_{L}$ within which the amount $x_{1}$ could vary was not significant before or after the addition of tenside. The only observable difference is that the settling time of $x_{1}$ was shortened due to the presence of tenside from cca 8 h to cca 5 h. This indicates that the tenside sped up the drug release and, in turn, the drug absorption as well.

On the other hand, as shown in Figure 2b and Figure 3b, the effect of the tenside on the amount $x_{2}$ in the central compartment was much more strongly manifested. The amount $x_{2}$ without tenside reached its peak value (cca 1.1 mg) 2 h after the administration, and this peak value may have ranged from cca 0.9 to 1.38 mg. Thus, the absolute range of the variation was 0.48 mg.

Contrary to this, the peak value of $x_{2}$ with tenside was much greater (cca 1.43 mg) and the possible range of its variations was approximately from 1.2 mg to 1.8 mg, that is, 0.6 mg in absolute terms; hence, the addition of tenside significantly increased not only the peak value, but also the range of its variations.

Within the experiments described above, only the effects of tenside on the dynamics of the drug amounts under the assumption of uncertain pharmacokinetics were monitored. This information is undoubtedly very useful for the drug designer on its own, but the upper and lower matrices $\underline{A}$, $\bar{A}$ and vectors $\underline{c}$, $\bar{c}$ can be much more effectively utilized after the augmentation of the uncertain model (5) by the state-bounding observer. In the next section, it is shown that such a combination may effectively work as a predictor of drug amounts in all compartments, while the performance of measurements is required in only one compartment.

### 2.4. State-Bounding Observer as a Predictor of Dangerous Therapy

In this section, it is proposed that the limiting amounts $x_{U}$ and $x_{L}$ are not generated according to Equation (5) for a certain Δ, but that these are predicted by the state-bounding observers. It is also suggested that this arrangement may significantly improve the safety of drug therapy.

This idea is demonstrated by the compartmental model, shown in Figure 4, which has three compartments: the GIT into which the drug is administered, the central compartment (the systemic blood circulation), and the tissue compartment from which the blood samples are typically collected.

Now, assume that blood samples are about to be collected repeatedly, but considering that the actions of various exogenous and endogenous effects cause the pharmacokinetic parameters to continually change over time. Therefore, it might be virtually impossible to receive unbiased information about the amount of drug in various body compartments. Obviously, a feasible approach is to augment the nominal pharmacokinetic model with the state-bounding observer, which is able to estimate and predict the drug amounts in all the compartments.

Assume that the same in vivo experiment as described above is performed with the same drug amounts measured, and consider again that there is a need to examine the effect of tenside on the drug amounts in the compartments. For a good illustration, the compartmental model shown in Figure 4 is used, while the blood samples are collected from the tissue compartment, i.e., from the system output $y=x_{3}$.

The model is described by the following set of differential Equations (8):(8)$$\left(\begin{array}{c} {\dot{x}}_{1}\left(t\right)\\ {\dot{x}}_{2}\left(t\right)\\ {\dot{x}}_{3}\left(t\right) \end{array}\right)=\left(\begin{array}{ccc} -\left(k_{a}+k_{e1}\right) & 0 & 0\\ k_{a} & -k_{23} & 0\\ 0 & k_{23} & -k_{e3} \end{array}\right)\left(\begin{array}{c} x_{1}\left(t\right)\\ x_{2}\left(t\right)\\ x_{3}\left(t\right) \end{array}\right)\;y\left(t\right)=\left(\begin{array}{ccc} 1 & 0 & 0 \end{array}\right)x\left(t\right)\;x\left(0\right)={\left(\begin{array}{ccc} 50 & 0 & 0 \end{array}\right)}^{T}\;\mathrm{mg}$$

The upper $x_{U}$ and lower $x_{L}$ limits of the drug amounts are estimated by two state-bounding observers in the form [5,12,13,14,15]:(9)$${\dot{x}}_{U}=\left(\bar{A}-L\underline{c}\right)x_{U}+Ly\;{\dot{x}}_{L}=\left(\underline{A}-L\bar{c}\right)x_{L}+Ly\;x\left(0\right)={\left(\begin{array}{ccc} 50 & 0 & 0 \end{array}\right)}^{T}\;\mathrm{mg}$$
where y is the output of the system (8) and L is the observer-gain vector.

Note that the dynamics of the compartmental model (8) are directly embedded into the observers (9) via the matrices $\underline{A}$, $\bar{A}$ and the vectors $\underline{c}$, $\bar{c}$. Both observers are driven by the output y(t) of the nominal model (8). The facts mentioned above are essential for further explanation. The arrangement of the whole system, i.e., the connection between the nominal model (8) and the state-bounding observer (9), is illustrated in Figure 5.

It should be remarked that Figure 5 does not illustrate any form of physical back-transport of the drug beyond the information-flow diagram. The observer only mathematically estimates and predicts the amount of the drug in the remaining compartments based on the measurements from the site of blood collection. We have to mention that the theory of state-bounding observers is beyond the scope of this paper; however, we stress that the drug amounts in every compartment must be non-negative and that the upper and lower limits $x_{U}$, $x_{L}$, must be non-negative as well. Accordingly, the matrices $\left(\bar{A}-L\underline{c}\right)$ and $\left(\underline{A}-L\bar{c}\right)$ must result in observers (9) that are positive systems as well [6].

## 3. Results

For the drug samples taken before the addition of the tenside (the third row in Table 1) and under the initial condition $x_{1}\left(0\right)=50\;\mathrm{mg}$, the following parameters of the pharmacokinetic model (8) were identified:(10)$$k_{a}=0.0370{\;h}^{-1}k_{e1}=0.1214{\;h}^{-1}k_{23}=1.2725{\;h}^{-1}\;k_{e3}=0.2171{\;h}^{-1}$$

It is worth noting that the absorption-rate constant $k_{a}$ has the same value as in the case of the two-compartment model originally identified in [1].

Assuming Δ=0.1, the following matrices A, $\bar{A}$, $\underline{A}$ were computed:(11)$$A=\left[\begin{array}{ccc} -0.1584 & 0 & 0\\ 0.0370 & -1.2725 & 0\\ 0 & 1.2725 & -0.2171 \end{array}\right]\;\bar{A}=\left[\begin{array}{ccc} -0.1426 & 0 & 0\\ 0.0407 & -1.1452 & 0\\ 0 & 1.3998 & -0.1954 \end{array}\right]\;\underline{A}=\left[\begin{array}{ccc} -0.1742 & 0 & 0\\ 0.0333 & -1.3998 & 0\\ 0 & 1.1452 & -0.2388 \end{array}\right]$$

After the addition of tenside (the fourth row in Table 1) the following model parameters were identified:(12)$$k_{a}=0.0438{\;h}^{-1}\;k_{e1}=0.9634{\;h}^{-1}k_{23}=0.2115{\;h}^{-1}\;k_{e3}=0.2438{\;h}^{-1}$$
and the corresponding matrices A, $\bar{A}$, $\underline{A}$ were computed:(13)$$A=\left[\begin{array}{ccc} -1.0072 & 0 & 0\\ 0.0438 & -0.2115 & 0\\ 0 & 0.2115 & -0.2438 \end{array}\right]\;\bar{A}=\left[\begin{array}{ccc} -0.9065 & 0 & 0\\ 0.0482 & -0.1903 & 0\\ 0 & 0.2326 & -0.2194 \end{array}\right]\;\underline{A}=\left[\begin{array}{ccc} -1.1079 & 0 & 0\\ 0.0394 & -0.2326 & 0\\ 0 & 0.1903 & -0.2682 \end{array}\right]$$

The evolution of the drug amounts in all the compartments together with the limiting curves $x_{U}$ and $x_{L}$ are shown in Figure 6.

## 4. Discussion

From Equation (9), it is evident that the state-bounding observers are driven by the output of the nominal model (8). However, since the nominal equations (8) are mathematical expressions describing the processes taking place in the real animal, they can be replaced by the real measurements, and the uncertainties hidden in the nominal model can be reflected in the observers. Therefore, instead of feeding the observers with the output of the simulation model, the observer can equally well utilize the real blood samples, or, more precisely, the smoothed series of in vivo samples collected from a body organ, as outlined in Figure 5.

In these arrangements, the observers work as predictors of the drug amounts in all the body compartments, which is of significant practical importance. Namely, the limiting curves $x_{U}$ and $x_{L}$ not only indicate the ranges of the possible variations in the drug amounts in all the body compartments, but also whether the drug amount in the central compartment may reach values out of the therapeutic range. This therapeutic range is depicted in Figure 6b by two horizontal lines. Using this strategy, it is possible to avoid the risk of the therapy becoming either ineffective or toxic. This kind of dangerous therapy may occur if a doctor administers the drug repeatedly and the pharmacokinetic parameters change between the individual doses.

The appropriateness of this research is substantiated by the ease of the software implementation of the state-bounding observer, and its utilization as an effective tool in the hands of a researcher, allowing the researcher to predict the intervals within which the drug amounts can be expected while collecting the blood samples from a single compartment only.

## 5. Conclusions

This paper presented promising solutions to two problems. The first consisted in the design of a generalized compartmental model with incorporated tuning parameter Δ, representing the size of the parametric uncertainties. Based on the chosen Δ, the maximal and minimal system matrices $\underline{A}$ and $\bar{A}$ were computed. Building on this, the generalized compartmental model was derived and simulated separately for $A=\underline{A}$ and $A=\bar{A}$, yielding the limiting curves of the drug amounts $x_{U}$ and $x_{L}$. These curves defined the intervals in which the drug amounts could range due to the uncertainty of the parameters. The practical application of the generalized model was demonstrated by the results obtained in the previously performed in vivo analysis of tenside effects (MLS) on the fate of a drug (sulfathiazole) in rats after oral administration. The results of the simulations convincingly showed that the presence of the tenside remarkably improved the absorption-rate constant, which in turn increased the drug amounts in the compartments.

Based on these results, a more complex problem with significant practical impact was resolved. It was supposed that the drug amounts (or possibly concentrations) were measured in a single part of the body (in a single body-compartment) and, based on this information, the amounts of the drug in the remaining parts of the body were predicted. This was possible thanks to a special system—the state-bounding observer, which was also designed. This observer works as a predictor of the drug amounts in all the compartments of the body, especially in those that are either inaccessible to measurements or from which blood collection would be difficult.

Knowledge about the upper and lower limits, $x_{U}$ and $x_{L}$, in the systemic circulation is essential to determine whether a drug amount or concentration will remain in the therapeutic range for a sufficiently long time, even if the characteristic parameters of the drug kinetics are not known with absolute certainty. This helps to prevent the therapy from becoming either ineffective or toxic. Fundamentally, the main contributions of this study involve the design of a state-bounding observer, which is directly driven by information on the drug amount collected from the peripheral blood rather than from the pharmacokinetic model.

The usefulness of the presented system-based approach lies in the possibility of predicting the fate of a drug in the body under various physiological and experimental conditions, which can significantly reduce the number of required in vivo experiments and, eventually, save many animals undergoing regular experiments. For instance, to the knowledge of the authors, there is no option other than the prediction of the drug amounts by a state-bounding observer if the drug must be administered repeatedly and the pharmacokinetic parameters vary between individual doses.

Finally, it is worth mentioning that the state-bounding observer is a mandatory means of designing control algorithms for general biosystems. Controlling the fate of a drug in the body is fundamentally based on optimizing a dosing regimen to keep the drug amounts (concentrations) at the desired levels for a sufficiently long time, despite uncertain pharmacokinetics and many adverse factors. These problems are the subject of current and future research by the authors.

## Figures and Tables

**Figure 1 pharmaceutics-14-00861-f001:**
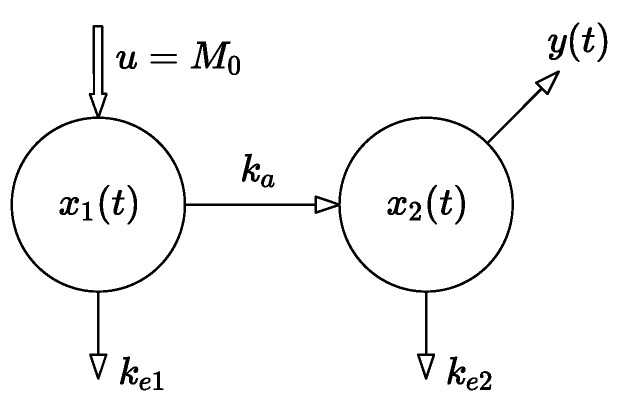
Structure of the two-compartment model of instantaneous oral administration [1].

**Figure 2 pharmaceutics-14-00861-f002:**
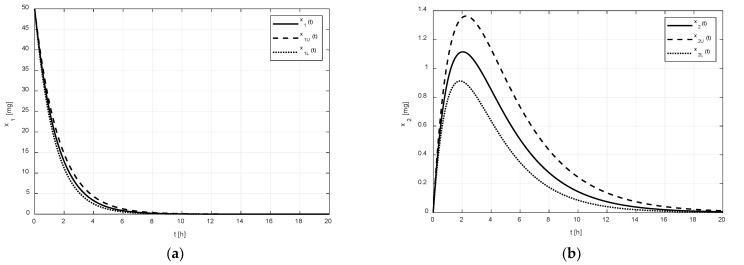
Regions in which amounts $x_{1}$ (**a**) and $x_{2}$ (**b**) can vary is delineated by the amount $x_{U}$ and $x_{L}$ of the drug without tenside.

**Figure 3 pharmaceutics-14-00861-f003:**
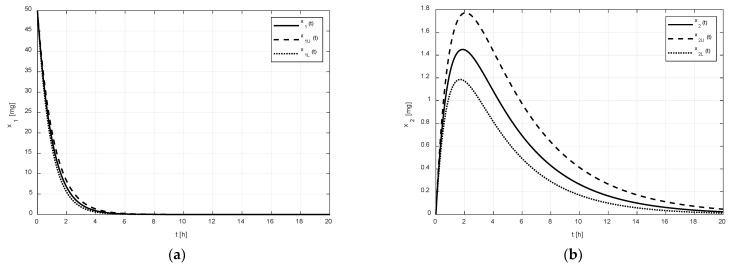
Regions in which amounts $x_{1}$ (**a**) and $x_{2}$ (**b**) can vary is delineated by the amount $x_{U}$ and $x_{L}$ of the drug with tenside.

**Figure 4 pharmaceutics-14-00861-f004:**
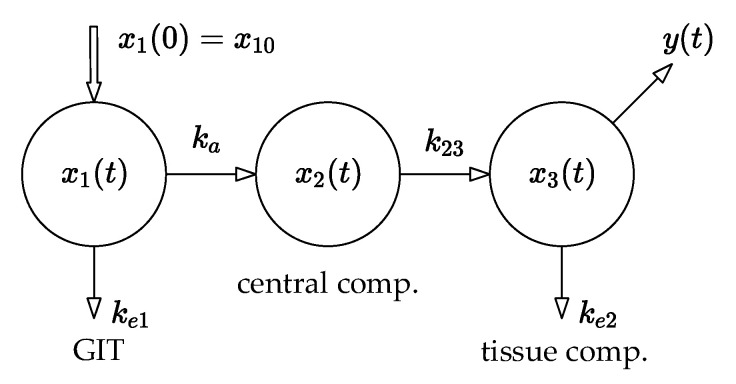
Three-compartment model of drug disposition.

**Figure 5 pharmaceutics-14-00861-f005:**
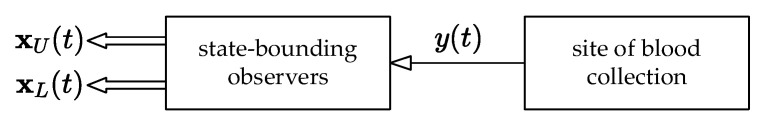
Practical use of the state-bounding observer.

**Figure 6 pharmaceutics-14-00861-f006:**
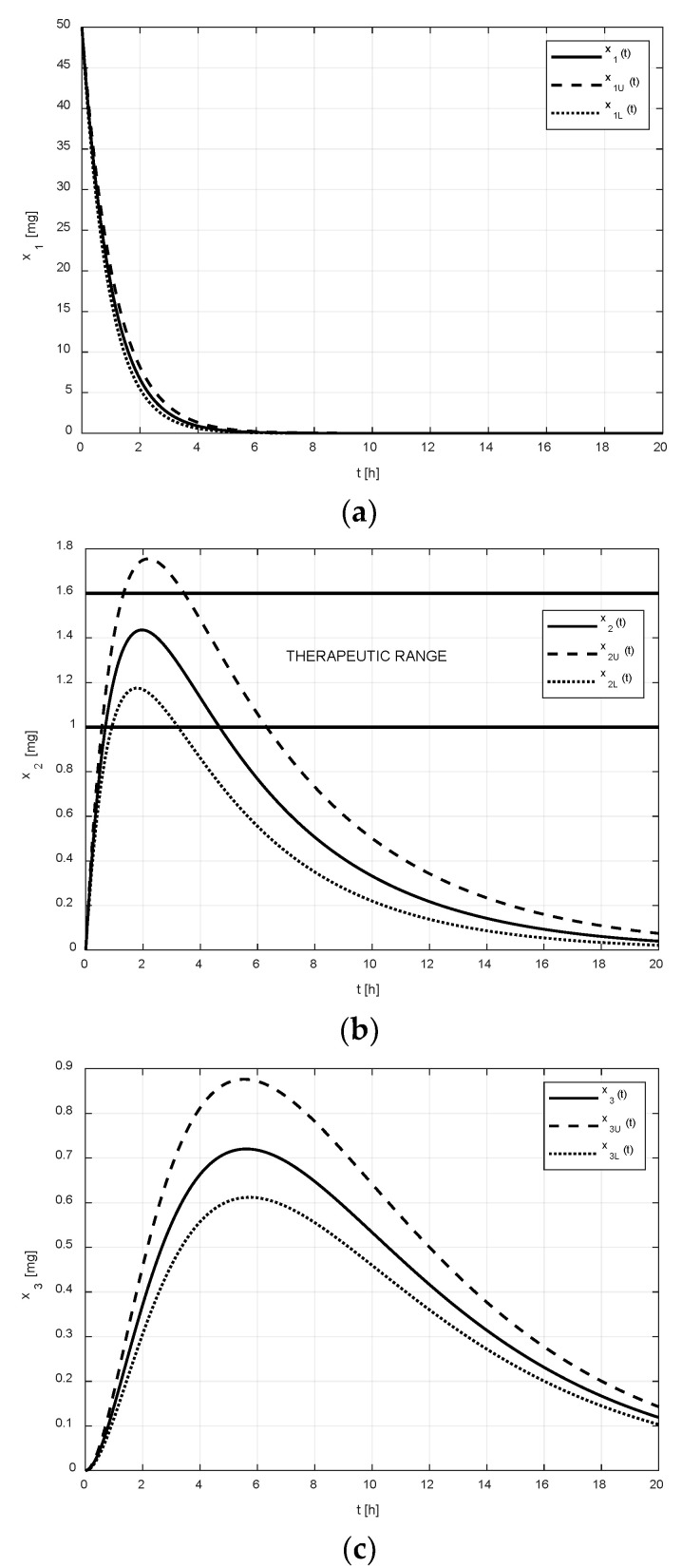
Regions in which the amounts $x_{1}\;$(**a**), $x_{2\;}$ (**b**), and $x_{3}\;$ (**c**) may potentially vary is delineated by the corresponding curves $x_{U}$ and $x_{L}$.

**Table 1 pharmaceutics-14-00861-t001:** In vivo concentrations and amounts of the drug without (x) and with (X) tenside [1].

*t* [h]	0	1	2	3	4	5	6
c (mg/mL)	0	0.0715	0.0855	0.0780	0.0735	0.0490	0.0535
x (mg)	0	0.914	1.093	0.997	0.952	0.626	0.684
X (mg)	0	1.262	1.403	1.280	1.222	0.803	0.878

## Data Availability

The in vivo data are available from the first author.

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
