# Peer review of "Using a State-Bounding Observer to Predict the Guaranteed Limits of Drug Amounts in Rats after Oral Administration Based on an Uncertain Pharmacokinetic Model"

_pharmaceutics, 2022, doi:10.3390/pharmaceutics14040861_

Round 1
Reviewer 1 Report
The manuscript describes an approach for solving the positive observation problem for linear positive systems. The outcome from pharmacokinetic (PK) studies is called uncertain but there are some PK models such as population pharmacokinetic models as well as physiologically based PK models that can explain inter-individual variability and to predict the drug disposition in precise way (Lines 140-146 – the sound of this paragraph is weak if the mentioned models are taken into account.). Therefore, I would suggest to talk about the application of state-bounding observer as additional tool for the refinement of PKs output. The manuscript is more suitable for a mathematical journal such as Applied Mathematics and Computation for example.
The manuscript has to be checked by a pharmacist or pharmacologist because it is not written in the language of pharmacology and it is difficult for reading because many untypical expressions exist. This is the main disadvantage of the manuscript. It cannot be published in its present form although the idea is good and from mathematical point of view is logical.
The introduction is very long.
The oral route is called as “per-oral”. The correct term is “oral administration”, “oral treatment” but not “per-oral”. The title has to be changed accordingly.
Bioavailability has a specific abbreviation in pharmacology and it is “F” and not “BA”.
MRT and MAT are not statistical parameters, their calculation is based on statistical moment theory.
“Exogenous influences” are called in the pharmacology literature as “factors affecting …………”
“Auxiliary substances” are “constitients”
Line 111: The sentence is not correct. It can be deleted. Because usually the blood samples is humans is obtained from the veins and from a finger.
The authors gave a scheme of the model on Figure 1 but it is strange that GItract is called peripheral compartment – it is the site of drug administration.
The model on Figure 4 does not acknowledged the back transport of the drug from tissue compartment to the central compartment. This brings some bias in the term of correct description of drug disposition in the body.
Line 275: Please, delete the second word “row”
Reviewer 2 Report
Zuzana Vitková etal., evaluated the work “Using state-bounding observer to predict guaranteed limits of 2 drug amounts in rats after per-oral administration based on uncertain pharmacokinetic model”
Below are the comments and provide the information in the manuscript:
- This manuscript contains different sections starting with the introduction, need for state-bounded observer, etc. and ends with conclusions. This manuscript doesn’t have material and methods, results, discussion sections. So it's confusing for the readers whether it is a research article or a review article.
- The authors need to check for similarity reports or plagiarism for the manuscript. There are some paragraphs (22-35) already available in the references included.
- There are many statements mentioned in the manuscript without any citations. The authors need to include references throughout the manuscript.
- In the 3.1 section, figure 1, table 1, equations 2a and 2b are already available in reference 1. The authors need to provide the reference number in the legends.
Reviewer 3 Report
Dear Authors,
I carefully read your paper that seemed promising.
I provided my corrections along the text using the pdf-comments.
I strongly encourage an English editing by a native speaker in order to permit a more fluent reading. Some topics are not well explained and a rephrasing is necessary (especially along the Introduction section).
I have to ask to the Editor a clarification about the format of this manuscript: it has been submitted as article but a section about Material and Methods is missing. Is it a review? A letter?
If the format of article is correct, I strongly encourage to insert the Material and Methods section in order to permit a better comprehension by the readers.

Round 2
Reviewer 1 Report
The authors significantly improved the readability of the manuscript. It has been revised and all the remarks were taken into account. Therefore, it can be accepted for publication.
Author Response
All spelling mistakes have been corrected.
Reviewer 2 Report
The authors significantly improved the manuscript and answered all the comments.
Spelling mistakes need to be corrected in the manuscript.
Author Response
All spelling mistakes have been corrected.
The appropriateness of the research has been justified and clarified in discussion
Reviewer 3 Report
Dear Authors, after the first round of revision you significantly improved the quality of the manuscript. I have no concerns about the approval. Have a good time.
Author Response

(The authors gave the same response as above.)
